# Association Between Temperature, Sunlight Hours, and Daily Steps in School-Aged Children over a 35-Week Period

**DOI:** 10.3390/jcm13247679

**Published:** 2024-12-17

**Authors:** Eva Rodríguez-Gutiérrez, Ana Torres-Costoso, Estela Jiménez-López, Arthur Eumann Mesas, Valentina Díaz-Goñi, María José Guzmán-Pavón, Nuria Beneit, Vicente Martínez-Vizcaíno

**Affiliations:** 1Health and Social Research Center, Universidad de Castilla-La Mancha, 16071 Cuenca, Spain; eva.rodriguez@uclm.es (E.R.-G.); estela.jimenezlopez@uclm.es (E.J.-L.); arthur.emesas@uclm.es (A.E.M.); valentina.diaz@uclm.es (V.D.-G.); profesor.nbeneit@uclm.es (N.B.); vicente.martinez@uclm.es (V.M.-V.); 2Research Network on Chronicity, Primary Care and Health Promotion (RICAPPS), 16071 Cuenca, Spain; 3Faculty of Physiotherapy and Nursing, Universidad de Castilla-La Mancha, 45071 Toledo, Spain; mariajose.guzman@uclm.es; 4Department of Psychiatry, Hospital Virgen de La Luz, 16071 Cuenca, Spain; 5Centro de Investigación Biomédica en Red de Salud Mental (CIBERSAM), Instituto de Salud Carlos III (ISCIII), 28029 Madrid, Spain; 6Postgraduate Program in Public Health, Universidade Estadual de Londrina, Londrina 86051-990, Brazil; 7ABC-Age Research Group, 16071 Cuenca, Spain; 8Facultad de Ciencias de La Salud, Universidad Autónoma de Chile, Talca 3465548, Chile

**Keywords:** children, sunlight, temperature, daily steps, physical activity, dose-response analysis

## Abstract

**Objective**: To examine the associations between gradients of average daily temperature and sunlight hours with daily steps over a 35-week period in school-aged children and to evaluate whether there were differences by sex. **Methods**: We conducted a follow-up study involving 655 children (50.8% girls, mean age 10.45 ± 0.95 years) from six public primary schools in Cuenca, Spain. We measured daily steps using Xiaomi Mi Band 3 Smart Bracelets (Xiaomi Corporation, Beijing, China) from October 2022 to June 2023 (over 35 weeks). We collected the average daily temperature from the local weather station in Cuenca and the sunlight hours during the same period. We used ANCOVA models and LOESS regression to examine the associations between gradients of average daily temperature and daily hours of sunlight with daily steps. Additionally, we performed a multiple linear regression model. **Results**: Our findings revealed significant variations in daily steps across the 35 weeks. The relationship between environmental factors and daily steps was non-linear in both girls and boys. The optimal values for higher activity levels were an average temperature of 14 °C and 13 h of sunlight. Furthermore, a 1 °C increase in temperature was associated with an increase of 74 ± 130 steps/day, while an increase of one hour of sunlight was associated with an increase of 315 ± 237 steps/day. However, the sunlight hours may act as a moderating factor. **Conclusions**: Our study showed a non-linear association between average daily temperature and the sunlight hours with daily steps over a 35-week period. Appropriate strategies may be needed to promote physical activity during periods of extreme temperatures or sunlight exposure.

## 1. Introduction

Physical activity (PA) is a crucial component of healthy development in school-aged children, contributing to improved physical fitness [1], mental well-being [2], and the prevention of non-communicable diseases, such as heart disease, type 2 diabetes, and cancers in adulthood [3]. However, PA levels among children have declined substantially in recent decades [4,5], raising global concern about the public health implications of this decline [6]. Physical inactivity has been observed to further increase as children enter adolescence, with PA levels decreasing at an average rate of 7% per year between the ages of 10 and 19 [7]. Understanding the determinants of daily PA is essential for designing effective interventions aimed at promoting active lifestyles among youth.

Environmental factors, particularly weather conditions, have been identified as significant influences on PA behaviors [8]. In this regard, the available evidence suggests that there is a seasonal variation in PA, with children being more active in the spring and summer months compared to the autumn and winter months [9]. Temperature, one of the most influential weather conditions, may also influence children’s willingness and ability to engage in outdoor activities, as demonstrated by a recent meta-analysis that reported positive associations between temperature and PA [8]. However, the findings are inconsistent, showing a curvilinear relationship between temperature with moderate-vigorous PA (MVPA) and total PA in children from southeast Australia and northeast America [10,11], respectively. In contrast, in northeast Spain and southwest France, higher temperatures have been linearly associated with significantly higher MVPA levels [12]. 

These inconsistencies highlight the potential influence of climatic variations on children’s engagement in PA and emphasize the need to consider geographical regions when interpreting research findings. These environmental differences have implications not only for children’s engagement in PA but also for the potential for differences in PA trends between countries. For example, children in colder winter climates, such as northeast America and northeast Spain, may have greater difficulties engaging in outdoor activities during this season. In contrast, those in regions with mild winters and warm summers, such as southeast Australia and southwest France, may have more opportunities for engagement throughout the year [13]. In the case of daylight hours, some studies reported higher overall PA during a longer duration of sunlight [10,14,15]. However, one study found negative associations [16]. 

Although there are other reliable tools for measuring PA, daily step counts have emerged as a practical and easily quantifiable metric for assessing children’s PA levels in both research and real-world settings [17,18]. The daily step count provides a comprehensive representation of the overall PA accumulated throughout the day. This is a crucial metric in studies examining the relationship between PA and environmental factors, as it offers a comprehensive insight into the PA patterns of an individual over a specified period. However, few studies have employed daily steps as an objective measure of PA [19,20]. 

Furthermore, most studies analyzing the determinants of PA, particularly daily steps, have been conducted over relatively short periods of time, often assessing PA for a period of seven days or less or comparing specific seasonal periods, which could contribute to bias because the data are collected in a restricted period of the year. Thus, there is limited evidence on how daily variations in temperature and sunlight hours impact daily steps in children over a prolonged period. This study addresses this gap by investigating how daily variations in temperature and sunlight influence PA over an academic year, offering a more comprehensive view of the impact of environmental factors on daily step counts. Furthermore, there is a paucity of evidence examining potential non-linear relationships between diverse temperature and sunlight thresholds and the number of daily steps taken by children, which could reveal how their PA is modified in response to different environmental conditions. 

Therefore, this study aimed to examine the associations across gradients of average daily temperature and sunlight hours with daily steps over a 35-week period in school-aged children and to evaluate whether there were differences by sex.

## 2. Methods

### 2.1. Study Design and Participants

Following the STROBE guidelines [21] (Appendix A), we performed prospective longitudinal analyses of data from the e-MOVI study, a study in which PA, dietary behavior, lifestyle, and cardiovascular risk were evaluated during childhood. Data collection began in October 2022 (yearly week 41) and ended in June 2023 (yearly week 23), lasting 35 weeks. Christmas holidays corresponded to week 52 and 1, and Easter Holidays corresponded to week 14. Each child was followed for 28 weeks, as data collection did not start at the same time for all subjects.

One thousand and forty-nine children aged 9 to 12 years, in 4th, 5th, or 6th grade, from 6 public primary schools in the province of Cuenca, Spain, were invited to participate in the research project. In order to participate in e-MOVI, the children had to meet the following criteria: (i) to be literate in Spanish (or Spanish sign language), (ii) to have no serious learning difficulties or physical and mental disorders that could prevent their participation, (iii) to have no allergies to the materials used in the smart wristband, and, (iv) and to have the consent from their parents or legal guardians to participate in the research project. A total of 745 (71.0%) schoolchildren were enrolled in the study. Of the total sample, 655 children (62.4%) were included in the present analyses, as 85 (0.8%) were excluded due to missing data on daily step counts for all weeks (Appendix A). 

### 2.2. Study Area

This study was conducted in the province of Cuenca, Spain, located in the central region of the Iberian Peninsula, the southernmost peninsula of the European continent. The villages considered for the study were Iniesta, Motilla del Palancar, San Clemente, Quintanar del Rey, Mota del Cuervo, and Las Pedroñeras (Appendix A). The main characteristics of the villages considered in the present study are shown in Appendix A.

Their location, altitude, and proximity to the sea result in a temperate continental Mediterranean climate, which is characterized by warm summers and cold winters [22]. Additionally, the duration of sunlight in Spain fluctuates throughout the year due to the country’s geographical location and seasonal changes. Spain is situated in the Central European Time Zone during winter and the Central European Sumer Time in summer. Its location in the southwestern region of Europe further influences the duration of sunlight throughout the year. The days are longer in the summer months (June to September) and shorter in the winter months (December to February). The longest day of the year occurs at the summer solstice, around 21 June, while the shortest day occurs at the winter solstice, around 21 December. In the study areas, the shortest day in 2022 had a mean of 9.35 h of sunlight, and the longest day in 2023 had a mean of 14.90 h of sunlight.

### 2.3. Study Variables

#### 2.3.1. Exposure: Environmental Factors

##### Daily Sunlight Hours

The mean daily sunlight hours were calculated for each week. Sunlight hours were defined as the time between sunrise and sunset. For each village, specific sunlight hours were determined.

##### Average Daily Temperature

The mean of the daily average temperature (°C) was calculated for each week. The daily average temperature was obtained from the meteorological data of the Spanish National Meteorological Agency (AEMET: https://www.aemet.es/, accessed on 27 February 2024) at the Cuenca weather station, located in the city of Cuenca at an altitude of 956 m above sea level. The mean distance from the participants’ villages to the weather station was 77.9 km (range from 59.1 km to 82.7 km).

#### 2.3.2. Outcome: Daily Steps

Participants wore a Xiaomi Mi Band 3 Smart Bracelet (Xiaomi Corporation, Beijing, China) on their non-dominant wrist to measure daily steps. This device has been validated for daily step counts [23]. 

Before initiating the study, the accuracy of the step-counting device was assessed on a subsample of the participants. This pilot calibration test was performed on a treadmill using a protocol that required participants to complete up to 10 sets lasting 3 min at 0% incline. Treadmill speed began at 0.8 km/h and increased by 0.8 km/h for each set. Participants were not given specific instructions on the type of locomotion to perform. The protocol ended when either the participant or the investigator decided to terminate the test. The accuracy of the device was assessed by comparing the recorded step count with a manual count. In addition, a video camera focused on the feet of each participant was used to obtain a secondary record. The intraclass correlation coefficient obtained was 0.93 between the step count recorded manually by direct observation and the count recorded by the Xiaomi Mi Band 3, indicating high agreement between the two methods.

The mean number of daily steps per week was calculated for participants whose wristbands contained data for at least 4 days per week, including at least 1 weekend day. The children recorded their daily steps, as displayed by the wristband, in a log, which was collected weekly at school by a member of the research team.

### 2.4. Statistical Analysis

Statistical (Kolmogorov–Smirnov test) and graphical methods (normal probability plots) were used to assess the normality of all continuous variables. The characteristics of the study sample were compared by sex using a Student’s *t*-test. Homogeneity of variances was assessed with the Levene test. For the remaining analyses, we used the 35 weeks of follow-up (from week 41, 2022, to week 23, 2023) as units of observation instead of the study subjects. Each week represented the mean of the schoolchildren for each of the variables analyzed, stratified by sex.

To explore the relationship between weeks of the year and daily steps, average daily temperature, and daily sunlight hours, we used a locally weighted scatterplot smoothing (LOESS) regression. We also used LOESS regression to examine the relationship among daily steps, daily hours of sunlight, and average daily temperature. 

We used analysis of covariance (ANCOVA) models to analyze the mean differences in daily steps as a dependent variable by average daily temperature tertiles (<8.8 °C, 8.8–13.7 °C, and >13.7 °C) and by daily sunlight hours tertiles (<10 h, 10–12 h, and >12 h). In Model 0, no adjustments were made, whereas in Model 1, we controlled for daily sunlight hours and average daily temperature, respectively. For significant associations, eta-squared values were also provided [24]. Additionally, we examined the interaction between average daily temperature and daily sunlight hours.

We performed a multiple linear regression model to estimate the linear association between average daily temperature and daily sunlight hours with daily steps (Model 0) and adjusted for average daily temperature and daily sunlight hours, respectively (Model 1). 

For subjects with missing daily step data, we performed data imputation to make the most of the available information and maintain statistical power. Multiple imputation is an approach used to compensate for missing data based on an automatic chained method selected through comprehensive data analysis. This involved generating five replications, which were subsequently pooled together in the analysis [25].

Analyses were conducted using the statistical software package IBM SPSS Statistics 29.0 (SPSS, Inc., Chicago, IL, USA) and JASP 0.18.3 software (University of Amsterdam, Amsterdam, The Netherlands). The statistical significance was set at two-tailed *p* < 0.05.

### 2.5. Ethics

The study protocol was approved by the Clinical Research Ethics Committee of the Hospital Virgen de la Luz de Cuenca (REG: 2019/PI1519). After approval by the Board of Directors of each school, a letter was sent to the parents of all grade-level students inviting them to a meeting. At the meeting, the study’s objectives were explained in detail, and parents were asked to provide written consent for their children’s participation. Furthermore, the study’s characteristics were explained to the schoolchildren, who provided their consent to participate. After data collection, parents were informed of their children’s results by letter. All procedures conducted in this study adhered to the Declaration of Helsinki and the Singapore Statement on Research Integrity [26].

## 3. Results

Table 1 shows the descriptive characteristics of the study participants (mean ± standard deviation). A total of 655 children were included in the study, with a mean age of 10.45 ± 0.95 years, of whom 333 (50.8%) were girls. There were statistically significant differences in daily steps between boys and girls. Appendix A presents the number of children followed up each week, along with the mean and standard deviation for daily steps, daily sunlight hours, and average daily temperature.

Figure 1 depicts the trend of the association between weeks of the year and daily steps, average daily temperature, and daily sunlight hours. In December, the number of daily steps and the number of sunlight hours reached their lowest point, and both increased from January onwards. However, there was a decline in daily steps from April (week 15, 12,416 ± 4820), while the number of sunlight hours continued to increase. Meanwhile, the temperature followed the same pattern as the sunlight hours, although it was lower in January and February. Consistently, daily steps exhibited a similar trend for both boys and girls throughout the year (Appendix A).

Figure 2 shows the nonlinear relationships between average daily temperature and daily sunlight hours with daily steps. However, the relationship between average daily temperature and daily sunlight hours was linear (Appendix A). The number of daily sunlight hours associated with the highest number of daily steps appears to be around 13 h for both girls and boys (Appendix A). Additionally, the number of daily steps increased as the average daily temperature was between 10 °C and 14 °C and decreased again from 14 °C onwards, except for boys, where it plateaued (Appendix A).

Based on the ANCOVA models, it was found that the number of daily steps taken by both girls and boys increased with an increase in average daily temperature (Table 2). However, significant differences were only observed between temperatures <8.8 °C and >13.7 °C for the total sample and in boys and between 8.8–13.7 °C and >13.7 °C in girls (Model 0). When the ANCOVA models were adjusted for daily sunlight hours (Model 1), the differences disappeared, except for girls, who were found to take fewer steps when the temperature was >13.7 °C.

Similarly, the number of daily steps taken by both girls and boys was found to increase with increasing daily sunlight hours, as indicated by ANCOVA models (Table 3). However, no significant difference was observed between 10–12 h and > 12 h of daily sunlight for boys (Model 0). This finding remained significant after controlling for average daily temperature (Model 1).

Furthermore, an interaction effect between daily sunlight hours and mean daily temperature (*p* < 0.001) was observed. Appendix A shows that there was no significant variation in the number of daily steps between temperature tertiles as a function of sunlight hours tertiles. Conversely (Appendix A), in each tertile of average daily temperature, the number of daily steps was higher when sunlight hours were higher. This was similar for girls and boys, except for the temperature tertile > 13.7 °C, where boys showed no difference in daily sunlight hours (Appendix A).

Finally, an increase of one hour of daily sunlight was associated with an increase of 315 ± 237 steps/day (416 ± 260 steps/day in girls and 235 ± 278 steps/day in boys; *p* < 0.001). Meanwhile, an increase of 1 °C of average daily temperature was associated with an increase of 74 ± 130 steps/day (86 ± 166 steps/day in girls and 69 ± 112 steps/day in boys; *p* < 0.05). However, after adjusting for daily sunlight hours and average daily temperature, respectively, the linear regression only remained significant for daily sunlight hours (Appendix A). 

## 4. Discussion

Our data revealed significant variations in daily steps over the 35 weeks in Spanish children. We observed that the relationship between daily steps and environmental factors was nonlinear in both girls and boys. The optimal values for higher activity levels were identified as an average temperature of 14 °C and 13 h of sunlight per day. However, the relationship between average temperature and daily steps was moderated by the number of sunlight hours. Furthermore, an increase of one hour of daily sunlight was associated with an increase of 315 ± 237 steps/day. 

The children’s daily step counts showed fluctuations across the academic year, exhibiting a pattern comparable to that observed in previous studies, with the highest levels of PA, including daily steps, in spring and the lowest in winter [9,27]. The observed seasonal variations in PA can be attributed to the environmental factors inherent to the respective seasons [28], since in spring and summer, for example, there is an increase in daylight hours and a rise in the average daily temperature compared to winter and autumn, which could make the practice of PA more comfortable. However, day-to-day variation in weather within seasons can also have an impact on PA. In this regard, our results from longitudinal data are consistent with those of previous studies, which reported higher levels of PA associated with higher temperatures [10,20,29] and longer periods of sunlight [10,14,15], including earlier research also conducted among Spanish children [12]. Nevertheless, the nonlinear relationships identified in our study indicate a decline in daily steps when an average temperature of 14 °C and 13 h of sunlight are reached. Similarly, in Australian and American children, the relationships between PA and temperature were curvilinear, and the optimal PA levels occurred at maximum temperatures between 20 °C and 25 °C [10,29] and at an average temperature of 20 °C [11], respectively. This suggests that the impact of temperature on PA is relative, with optimal ranges varying depending on climate. These findings emphasize the importance of contextual factors when assessing the relationship between environmental factors and PA levels.

The amount of time spent outdoors has been identified as a key predictor of overall PA levels in children, as it is generally spent in more physically active behaviors than time spent indoors [30]. Children are more active when the weather is more conducive to outdoor activity [13]. The observed plateau in boys or decline in girls in activity at higher temperatures observed in our cohort may be indicative of a threshold beyond which children, particularly girls, reduce their outdoor activity due to discomfort or safety concerns related to excessive heat. This sex difference could be due to differences in daily step levels between boys and girls, as our data showed that girls walked, on average, 2217 steps/day less than boys. These differences may be due to a complex interplay of cultural and social factors. Boys are more likely to engage in sporting activities for longer periods of time and more often than girls due to their greater inclination toward activities that emphasize physical fitness, such as team sports. Conversely, social interaction is a more important motivating factor for girls [31,32,33]. Additionally, the PA habits of both parents have a notable impact on girls’ participation in PA, whereas only the PA habits of the father are associated with the amount of time boys spend in PA [33].

The reduction in daily steps on days with > 13 h of sunlight may be attributed to children engaging in other activities that do not necessarily involve movement, such as social activities, the use of electronic devices, or academic-related activities. The findings of Ren et al. [34] indicate that the increase in sedentary time can be attributed to excessive homework during weekdays. In our case, the days of maximum sunlight hours coincided with the end of the school year, when final exams are concentrated, leading to an increase in study time. However, these hypotheses require confirmation and additional research to elucidate the underlying mechanisms responsible for this inverted U-shaped relationship. However, there are notable differences between a child’s typical daily routine during the academic year and that of non-school days, including weekends and holidays. Previous research suggests that structured environments, such as school days, are associated with higher levels of PA due to the inclusion of scheduled physical education classes, recess periods, and after-school activities. In contrast, non-school days may be characterized as less formally structured environments [35].

Notably, we found a significant interaction between sunlight hours and temperature for the mean difference in daily steps. It was observed that children accumulated a greater number of daily steps on days with a higher number of sunlight hours, regardless of the average temperature. This suggests that sunlight hours may act as a moderating factor in the relationship between temperature and daily steps. This finding corroborates the hypothesis proposed by Beighle and colleagues [28], suggesting that children may remain active in colder climates if they have access to longer daylight hours simply because they have more hours to be outside. However, contrary to our findings, Duncan et al. [19] observed that the relationship between temperature and daily steps was independent of day length in New Zealand children. In contrast, the association of day length with daily steps was unclear after adjustment for the temperature. These observed differences may be attributed to variations in the geographical context or the duration of the follow-up period, given that the study was conducted between August and December (winter to summer). 

By examining both seasonal variations and sex-specific responses to environmental conditions, this study offers valuable insights into the influence of weather on daily step counts in school-aged children. The findings from this study emphasize the importance of assessing PA over extended periods, since short-term studies or those focused on a specific season may not capture the full complexity of the variability in children’s activity levels in response to environmental factors. By examining these associations over a 35-week period, our study provides a more comprehensive understanding of the impact of daily variations in temperature and sunlight on PA. This long-term perspective is crucial for the development of PA programs that are effective throughout the entire school year rather than only during optimal weather conditions. The findings also emphasize the importance of optimizing opportunities for outdoor activity on days with favorable weather conditions. Furthermore, additional strategies may be necessary to encourage increased PA during periods of extreme temperatures or during periods of shorter daylight hours. As higher temperatures may discourage activity, especially among girls, promoting shaded areas or scheduling outdoor activities at cooler times of the day (such as early mornings or late afternoons) could be beneficial. During colder months with fewer sunlight hours, organized indoor PA programs, such as sports, dance, and gym-based free play, can help maintain activity levels in comfortable settings.

### Limitations and Strengths

The study has several limitations that must be acknowledged. Firstly, the observational nature of the study limits our ability to infer causality between environmental factors and PA. Although we observed significant associations, we could not definitively conclude that changes in temperature and sunlight hours caused changes in daily steps. Experimental studies are needed to establish causal relationships. Secondly, we included daily variations in temperature and sunlight hours due to their established relevance in the literature and the feasibility of data collection during the study period. However, other environmental factors, such as humidity, wind speed, and air quality, were not included in the analysis. Thirdly, the study was conducted in a specific geographical region (Spain), which may limit the generalisability of the findings to other climates or regions. Different regions may experience different environmental influences, and thus, caution should be exercised when applying these results to populations in other parts of the world with distinct climatic conditions. Fourthly, the long-term data collection period included school and holiday periods, which could lead to variability in daily step counts due to changes in children’s daily schedules. It would be beneficial for future studies to consider school and holiday periods as covariates to accurately capture these influences. Fifthly, although the Xiaomi Mi Band 3 has been validated in a sample of Spanish participants in free-living conditions [23], the time of year was not specified, and weather conditions were not controlled during data collection. Future studies should consider validating the device in different climatic contexts to ensure consistency and accuracy of results in different environmental conditions. Finally, the data on daily steps were self-reported by the children, which could introduce the potential for reporting bias. However, we were unable to access the direct data from the schoolchildren due to Xiaomi Inc.’s privacy policy, which prevented access to the data. Thus, we endeavored to minimize bias and enhance reliability by implementing a rigorous data collection procedure and providing clear instructions to the participants. Future studies should consider using objective measures to improve data reliability.

Despite its limitations, our study has several strengths that enhance the validity and generalisability of its findings. Firstly, the longitudinal design, which encompasses a period of 35 weeks, offers a comprehensive insight into the fluctuations in daily PA, as measured by step counts, in response to environmental variables over time. By collecting data across multiple seasons, we were able to mitigate the biases associated with seasonal variations that are commonly observed in studies with shorter durations. Secondly, the large sample size of 655 children allows for robust analysis and the ability to detect meaningful differences in activity patterns, including sex-based variations.

## 5. Conclusions

Our study found that daily steps accumulated by both girls and boys demonstrated a positive association with daily sunlight hours and average daily temperature over a 35-week period. However, these relationships were nonlinear, and the highest number of steps was observed at an average temperature of 14 °C and 13 h of sunlight. A significant interaction between these two factors was observed, indicating that the relationship between average temperature and daily steps is moderated by the number of sunlight hours. These results highlight the need for tailored strategies to maintain or even enhance PA during periods of extreme temperatures or sunlight exposure, ensuring that children remain active regardless of the season. Further research is needed to assess how children’s use of leisure time changes under varying environmental conditions.

## Figures and Tables

**Figure 1 jcm-13-07679-f001:**
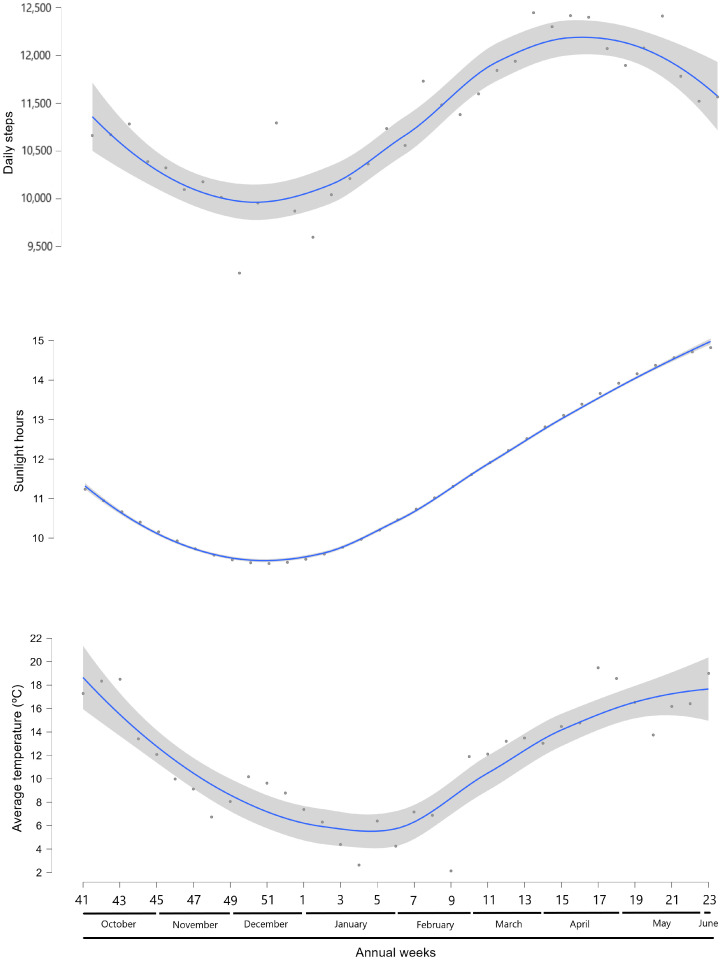
Scatterplots illustrating LOESS regression analysis between weeks of the year (from week 41, 2022, to week 23, 2023) and daily steps, average daily temperature (°C), and daily sunlight hours. Complete n (*n* = 655): week 49 to week 17. Christmas holidays correspond to weeks 52 and 1. Easter holidays correspond to week 14.

**Figure 2 jcm-13-07679-f002:**
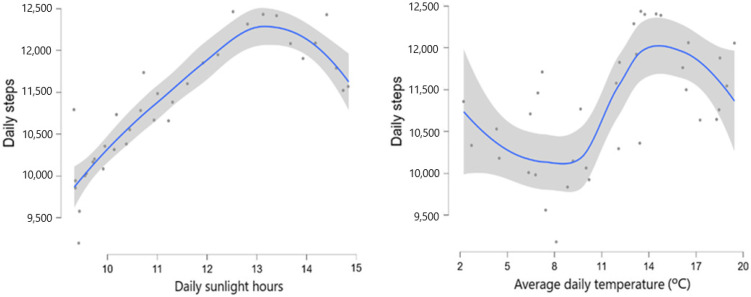
Scatterplots illustrating LOESS regression analysis between daily steps and average daily temperature (°C) and daily sunlight hours.

**Table 1 jcm-13-07679-t001:** Characteristics of the study sample and environmental conditions.

	Total (*n* = 655)	Girls (*n* = 333)	Boys (*n* = 322)	*p*-Value
Age (year)	10.45 ± 0.95	10.51 ± 0.96	10.38 ± 0.94	0.088
Weight (kg)	39.22 ± 10.36	39.71 ± 10.63	38.71 ± 10.07	0.217
Body mass index, kg/m^2^	18.85 ± 3.76	18.89 ± 3.77	18.80 ± 3.75	0.757
Daily steps	11,279 ± 3029	10,189 ± 2631	12,406 ± 3006	**<0.001**
Average weekly temperature (°C)	11.50 ± 4.98	11.50 ± 4.98	11.50 ± 4.98	**-**
Daily sunlight hours	11.44 ± 1.83	11.44 ± 1.83	11.44 ± 1.83	**-**

Values are mean ± standard deviation. The values in bold indicate statistical significance at *p* < 0.05.

**Table 2 jcm-13-07679-t002:** Analysis of covariance of daily steps (total sample, girls, and boys) by average daily temperature (°C) categories over 35 weeks.

		Average Daily Temperature Categories			*p*-Value	Eta Square
		Low (L) <8.8 °C	Medium (M) 8.8–13.7 °C	High (H) >13.7 °C		
	n Weeks	12	11	12		
Daily steps total sample	M_0_	10,810 ± 597 ^H^	11,352 ± 712	11,813 ± 475 ^L^	**<0.001**	0.346
	M_1_	11,295 ± 509	11,541 ± 434	11,154 ± 575	0.159	0.112
Daily steps girls	M_0_	9643 ± 547 ^H^	10,233 ± 907	10,840 ± 785 ^L^	**0.002**	0.319
	M_1_	10,345 ± 527	10,507 ± 448 ^H^	9888 ± 592 ^M^	**0.042**	0.185
Daily steps boys	M_0_	12,016 ± 657 ^H^	12,503 ± 568	12,892 ± 384 ^L^	**0.002**	0.325
	M_1_	12,294 ± 603	12,611 ± 514	12,515 ± 679	0.352	0.065

Data are presented as mean ± standard deviation (SD). The values in bold indicate statistical significance at *p* < 0.05. Model 0 (M_0_): raw data analysis. Model 1 (M_1_): controlling for daily sunlight hours. The superscript letters indicate statistical significance (*p* < 0.05) between categories for post-hoc tests using the Bonferroni comparisons. Eta squared values of 0.01, 0.06, and 0.14 indicate small, intermediate, or strong effect sizes, respectively.

**Table 3 jcm-13-07679-t003:** Analysis of covariance of daily steps (total sample, girls, and boys) by daily sunlight hours categories over 35 weeks.

Daily Sunlight Hours Categories						
		Low (L) <10 h	Medium (M) 10–12 h	High (H) >12 h	*p*-Value	Eta Square
	n Weeks	11	12	12		
Daily steps total sample	M_0_	10,533 ± 403 ^M,H^	11,306 ± 317 ^L,H^	12,069 ± 334 ^L,M^	**<0.001**	0.774
	M_1_	10,446 ± 408 ^M,H^	11,292 ± 350 ^L,H^	12,162 ± 423 ^L,M^	**<0.001**	0.710
Daily steps girls	M_0_	9354 ± 295 ^M,H^	10,036 ± 483 ^L,H^	11,254 ± 422 ^L,M^	**<0.001**	0.799
	M_1_	9185 ± 451 ^M,H^	10,010 ± 385 ^L,H^	11,436 ± 468 ^L,M^	**<0.001**	0.779
Daily steps boys	M_0_	11,748 ± 523 ^M,H^	12,636 ± 275 ^L^	12,963 ± 389 ^L^	**<0.001**	0.632
	M_1_	11,775 ± 481 ^M,H^	12,640 ± 412 ^L^	12,935 ± 499 ^L^	**<0.001**	0.489

Data are presented as mean ± standard deviation. The values in bold indicate statistical significance at *p* < 0.05. Model 0 (M_0_): raw data analysis. Model 1 (M_1_): controlling for average daily temperature (°C). The superscript letters indicate statistical significance (*p* < 0.05) between categories for post-hoc tests using the Bonferroni comparisons. Eta-squared values of 0.01, 0.06, and 0.14 indicate small, intermediate, or strong effect sizes, respectively.

## Data Availability

The datasets used and/or analyzed during the current study are available from the corresponding author upon reasonable request.

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
