# Peer review of "Association Between Temperature, Sunlight Hours, and Daily Steps in School-Aged Children over a 35-Week Period"

_jcm, 2024, doi:10.3390/jcm13247679_

Round 1
Reviewer 1 Report
Comments and Suggestions for Authors
1. There is a high percentage of overlap in the manuscript.
2. The introduction has to be improved, adding novelty of the study.
3. The introduction needs to be more critical . For eg: The authors are explaining the PA trends in various countries . what is the is the difference between these countries in terms of PA and environment and what difference is there with the current population has to be described in the introduction . Why cant those findings cannot be generalized to the current population has to be explained .
4. Why did the authors opt for daily steps as an outcome measure? It cannot measure the intensity of the activity
5. There are other confounding factors such as humidity, extreme weather condition, air quality, wind which may influence the PA . How the authors addressed these variables in the research
6. Please provide a detailed flow chart on the participant's recruitment and participant flow
7. What is the validity of the tool used for step count in the age group studied? Is it accurate in various weather conditions?
8. How was the device calibration performed?
9. The recording of the data is by self reporting .This may cause bias in reporting as the participants are young children
10. The table shows a gender-based difference in daily steps and environmental response. This part has to be further explored
11. The discussion has to be improved.There is no interpretation or critical analysis of the result in the discussion section
12. What is the practical implication of this study?
13. Add recommendation
Author Response
Comments 1:
- There is a high percentage of overlap in the manuscript.
Response 1:
Thank you for the reviewer's comment. We modified the manuscript to make sure there is no overlap.
Comments 2:
- The introduction has to be improved, adding novelty of the study.
Response 2:
Thank you for the reviewer's comment. We have revised the introduction to provide a fuller explanation of the study's novelty as follows:
Introduction section:
“This study addresses this gap by investigating how daily variations in temperature and sunlight influence PA over an academic year, offering a more comprehensive view of the impact of environmental factors on daily step counts. Furthermore, there is a paucity of evidence examining potential non-linear relationships between diverse temperature and sunlight thresholds and the number of daily steps taken by children, which could reveal how their PA is modified in response to different environmental conditions.”
Comments 3:
- The introduction needs to be more critical . For eg: The authors are explaining the PA trends in various countries . what is the is the difference between these countries in terms of PA and environment and what difference is there with the current population has to be described in the introduction . Why cant those findings cannot be generalized to the current population has to be explained .
Response 3:
Thank you to the reviewer for his/her thoughtful comment. The reviewer's recommendations for refinement have been carefully considered and incorporated into our revised manuscript.
Introduction section:
“These inconsistencies highlight the potential influence of climatic variations on children's engagement in PA and emphasize the need to consider geographical regions when interpreting research findings. These environmental differences have implications not only on children's engagement in PA but also on the potential for differences in PA trends between countries. For example, children in colder climates may have greater difficulties in engaging in outdoor activities during the winter months, whereas those in milder regions may have more opportunities for engagement throughout the year [13].”
Comments 4:
- Why did the authors opt for daily steps as an outcome measure? It cannot measure the intensity of the activity
Response 4:
We appreciate the reviewer's comment. The selection of daily steps as an outcome measure is based on its practicality and the fact that it represents an accessible and easily understandable indicator of daily physical activity, particularly in child populations. While it is true that daily steps do not measure activity intensity, they do capture overall physical activity accumulated throughout the day, which is relevant in studies that observe trends in physical activity according to environmental factors, such as temperature and hours of sunlight, main objective of this study.
Comments 5:
- There are other confounding factors such as humidity, extreme weather condition, air quality, wind which may influence the PA . How the authors addressed these variables in the research
Response 5:
Thanks to the reviewer for this thoughtful comment. It is acknowledged that factors such as humidity, extreme weather conditions, wind, and air quality (particularly, outdoor airborne allergens) can also exert an influence on levels of physical activity. In this study, however, we focused on temperature and hours of sunlight as the main environmental variables. Thus, we have included a sentence acknowledging this limitation as follows:
Limitations and strengths section:
“Secondly, we included daily variations in temperature and sunlight hours due to their established relevance in the literature and the feasibility of data collection during the study period. However, other environmental factors, such as humidity, wind speed, and air quality, were not included in the analysis.”
Comments 6:
- Please provide a detailed flow chart on the participant's recruitment and participant flow
Response 6:
Thank you for the reviewer´s comment. Figure S1 presents a flow chart which illustrates that 1042 children were invited to participate in the project. Of these, 304 actively declined or did not return the written approval. Consequently, 745 children were included in the project, of whom 655 were included in the present analyses.
Comments 7:
- What is the validity of the tool used for step count in the age group studied? Is it accurate in various weather conditions?
Response 7:
Thank you for the comment. The Xiaomi Mi Band 3 is valid for measuring the number of steps in unstructured free-living conditions. In addition, it accurately classifies subjects according to whether or not they meet the 10,000 steps/day recommendations. The validation of this device was performed on Spanish participants, which reinforces the relevance of the data for our target population (1). However, we acknowledge the lack of specific studies exploring the accuracy of this device in different climatic conditions or at different times of the year. This represents a limitation of the study, as it acknowledges the lack of previous research in this area. It also serves to suggest this as a topic for further investigation in future studies.
References:
- Casado-Robles, C.; Mayorga-Vega, D.; Guijarro-Romero, S.; Viciana, J. Validity of the Xiaomi Mi Band 2, 3, 4 and 5 Wristbands for Assessing Physical Activity in 12-to-18-Year-Old Adolescents under Unstructured Free-Living Conditions. Fit-Person Study. J Sports Sci Med 2023, 22, 196, doi:10.52082/JSSM.2023.196.
Limitations and strengths section:
“Fifthly, although the Xiaomi Mi Band 3 has been validated on a sample of Spanish participants in free-living conditions [23], the time of year was not specified, and weather conditions were not controlled during data collection. Future studies should consider validating the device in different climatic contexts to ensure consistency and accuracy of results in different environmental conditions.”
Comments 8:
- How was the device calibration performed?
Response 8:
Thank you for the reviewer´s comment. Before initiating the study, the accuracy of the step-counting device was assessed on a subsample of the participants. This pilot calibration test was performed on a treadmill using a protocol that required participants to complete up to 10 sets lasting 3-min at 0% incline. Treadmill speed was started at 0.8 km/h and increased 0.8 km/h for each set. Participants were not given specific instructions on the type of locomotion to perform. The protocol ended when either the participant or the investigator decided to terminate the test. The accuracy of the device was assessed by comparing the recorded step count with a manual count. In addition, a video camera focused on the feet of each participant was used to obtain a secondary record. The intraclass correlation coefficient obtained was 0.93 between the step count recorded manually by direct observation and the count recorded by the Xiaomi Mi Band 3, indicating high agreement between the two methods.
Comments 9:
- The recording of the data is by self reporting. This may cause bias in reporting as the participants are young children
Response 9:
The reviewer’s comment seems judicious. This is acknowledged and reflected in the limitations of the study.
Limitations and strengths section:
“Finally, the data on daily steps were self-reported by the children, which could introduce the potential for reporting bias. However, we endeavoured to minimise bias and enhance reliability by implementing a rigorous data collection procedure and providing clear instructions to the participants. Future studies should consider using objective measures to improve data reliability.”
Comments 10:
- The table shows a gender-based difference in daily steps and environmental response. This part has to be further explored
Response 10:
Thank you for your observation. We agree that the gender-based differences in daily steps and responses to environmental factors deserve further exploration. To address this, we expanded our discussion to consider possible explanations for these differences.
Discussion section:
“The observed plateau in boys or decline in girls in activity at higher temperatures observed in our cohort may be indicative of a threshold beyond which children, particularly girls, reduce their outdoor activity due to discomfort or safety concerns related to excessive heat. This sex difference could be due to differences in daily steps levels between boys and girls, as our data showed that girls walked on average 2,217 steps/day less than boys. These differences may be due to a complex interplay of cultural and social factors. Boys are more likely to engage in sporting activities for longer periods of time and more often than girls, due to their greater inclination towards activities that emphasise physical fitness, such as team sports. Conversely, social interaction is a more important motivating factor for girls [31–33]. Additionally, the PA habits of both parents have a notable impact on girls' participation in PA, whereas only the PA habits of the father are associated with the amount of time boys spend in PA [33].”
Comments 11:
- The discussion has to be improved. There is no interpretation or critical analysis of the result in the discussion section
Response 11:
Thank you for the comment. As the reviewer suggests, we have added some extra information in the discussion section, as follows:
Discussion section:
“This suggests that the impact of temperature on PA is relative, with optimal ranges varying depending on climate. This emphasises the importance of contextual factors when assessing the relationship between environmental factors and PA levels.”
“This sex difference could be due to differences in daily steps levels between boys and girls, as our data showed that girls walked on average 2,217 steps/day less than boys. These differences can be attributed to a complex interplay of cultural and social factors. Boys are more likely to engage in sporting activities for longer periods of time and with greater frequency than girls, due to their greater inclination towards activities that emphasise physical fitness, such as team sports. Conversely, social interaction serves as a more significant motivating factor for girls [31–33]. Additionally, the PA habits of both parents have a notable impact on girls' participation in PA, whereas only the PA habits of the father are associated with the amount of time spent on PA by boys [33].”
“However, there are notable differences between a child's typical daily routine during the academic year and that of non-school days, including weekends and holidays. Previous research suggests that structured environments, such as school days, are associated with higher levels of PA due to the inclusion of scheduled physical education classes, recess periods, and after-school activities. In contrast, non-school days may be characterised as a less formally structured environment [35].”
“By examining both seasonal variations and sex-specific responses to environmental conditions, this study offers valuable insights into the influence of weather on daily step counts in school-aged children. […] The findings also emphasise the importance of optimising opportunities for outdoor activity on days with favourable weather conditions. Furthermore, additional strategies may be necessary to encourage increased PA during periods of extreme temperatures or during periods of shorter sunlight. As higher temperatures may discourage activity, especially among girls, promoting shaded areas or scheduling outdoor activities at cooler times of day (such as early mornings or late afternoons) could be beneficial. During colder months with fewer sunlight hours, organised indoor PA programmes, such as sports, dance, and gym-based free play, can help maintain activity levels in comfortable settings.”
Comments 12:
- What is the practical implication of this study?
Response 12:
We appreciate the comment. As the reviewer suggests we have added a recommendation, as follows:
Discussion section:
“By examining both seasonal variations and sex-specific responses to environmental conditions, this study offers valuable insights into the influence of weather on daily step counts in school-aged children. […] The findings also emphasise the importance of optimising opportunities for outdoor activity on days with favourable weather conditions. Furthermore, additional strategies may be necessary to encourage increased PA during periods of extreme temperatures or during periods of shorter sunlight. As higher temperatures may discourage activity, especially among girls, promoting shaded areas or scheduling outdoor activities at cooler times of day (such as early mornings or late afternoons) could be beneficial. During colder months with fewer sunlight hours, organised indoor PA programmes, such as sports, dance, and gym-based free play, can help maintain activity levels in comfortable settings.”
Comments 13:
- Add recommendation
Response 13:
We appreciate the reviewer's comments.
Discussion section:
“The findings also emphasise the importance of optimising opportunities for outdoor activity on days with favourable weather conditions. […] Since higher temperatures may discourage activity, especially among girls, promoting shaded areas or scheduling outdoor activities in cooler parts of the day (such as early mornings or late afternoons) could be beneficial. During colder months with fewer sunlight hours, organised indoor PA programmes, such as sports, dance, and gym-based free play, can help maintain activity levels in comfortable settings.”
Reviewer 2 Report
Comments and Suggestions for Authors
The study investigated the potential relationship between daily temperature and sunlight hours with daily steps across a 35-week period in school-aged children (boys and girls). The study indicated that, over the course of a 35-week period, there was a correlation between the number of steps taken by both girls and boys and the amount of sunlight and average temperature. However, it should be noted that these relationships were non-linear, and the highest number of steps was observed at an average temperature of 14°C and 13 hours of sunlight.
Regarding formal aspects, the manuscript is well structured and contains the following sections: Introduction; Materials and Methods; Results; Discussion; Strengths and limitations ending with Conclusions. The title is short, sufficiently specific and descriptive of the work. The language is appropriate, the text is clear, precise and objective. There is no evidence of overlap in the manuscript.
Abstract: It is appropriate (objective, methods, main results, and conclusion) and reflects the content of the manuscript.
L21 - “the dose-response” does not send appropriate term for a descriptive study, it refers for Experimental studies
L26 – between t avagere daily temperature?
Introduction: The research problem is clearly stated and defined. It is appropriately contextualized in relation to existing knowledge, moving from the general to the specific. The reasons justifying the need for the study are well established in the text. The references used to support the presentation of the research problem are current and relevant to the topic. The objective is clearly stated.
L74 – “the dose-response”
Materials and Methods: The methodological procedures are generally appropriate to the study of the research question and are sufficiently detailed and described in an adequate, clear and objective manner.
L146-152 – Why did you categorize daily sunlight hours tertiles and daily temperature tertiles?
L153 -155 - This explanation about the adjustment of model 0 and model 1 is confusing
Results: Presents the results in an appropriate manner, highlighting the main findings and avoiding unnecessary repetition. Make appropriate use of tables and figures and facilitate appropriate dissemination of results.
Table 1. - it doesn't seem appropriate to me to say that it is “Average weekly temperature (ºC) exposure” and “Daily sunlight hours exposure”. The word exposure sounds as if everyone was exposed to it in the same way, which is impossible, even for social reasons, preferences, social commitments, school... even though the week and day had such climatic conditions. I suggest just eliminating the word "exposure"
L216- “no significant difference was observed between 10-12 hours and > 12 hours of daily sunlight for boys (model 0).” However, Table 3 Superscript letter indicates statistical significance?
Discussion Presents a discussion of the limitations of the study and of existing knowledge on the topic. The strengths of the study are presented and discussed, suggesting the potential contributions of the main findings of the study to scientific development, innovation, or intervention in practice.
The level of physical activity is multifactorial, and in the case of children, the school period (vacation days or classes) will have a direct influence. You mentioned something in lines 243 and 270-272. This is a factor that should be considered as a covariate in future studies. But it does not invalidate this study. I just think it should be further explored in the discussions and explored in the limitations of the study. Additionally, the methodology, and figures 1 and S3, could somehow indicate the weeks in which the children were in school or on vacation weeks.
Supplementary material: Good quality material that facilitates the graphical presentation of the main findings of the study.
References: The references are updated and sufficient, most of them are made up of references from original articles and the citations in the text are adequate, that is, the statements in the text cite references that substantiate such statements.
Author Response
Comments 1:
The study investigated the potential relationship between daily temperature and sunlight hours with daily steps across a 35-week period in school-aged children (boys and girls). The study indicated that, over the course of a 35-week period, there was a correlation between the number of steps taken by both girls and boys and the amount of sunlight and average temperature. However, it should be noted that these relationships were non-linear, and the highest number of steps was observed at an average temperature of 14°C and 13 hours of sunlight.
Response 1:
Thank you for the reviewer´s comment. We appreciate the reviewer´s effort and comments on the manuscript.
Comments 2:
Regarding formal aspects, the manuscript is well structured and contains the following sections: Introduction; Materials and Methods; Results; Discussion; Strengths and limitations ending with Conclusions. The title is short, sufficiently specific and descriptive of the work. The language is appropriate, the text is clear, precise and objective. There is no evidence of overlap in the manuscript.
Response 2:
Thank you for the reviewer’s comment.
Comments 3:
Abstract: It is appropriate (objective, methods, main results, and conclusion) and reflects the content of the manuscript.
Response 3:
Thank you for the reviewer´s comment.
Comments 4:
L21 - “the dose-response” does not send appropriate term for a descriptive study, it refers for Experimental studies
Response 4:
We appreciate the reviewer's thoughtful consideration of the terminology employed in our study, given that it is a descriptive study. In accordance with the reviewer's suggestion, the term 'dose-response analysis' has been modified throughout the manuscript.
Abstract section:
“Objective: To examine the associations across gradients of average daily temperature and sunlight hours with daily steps across […].”
“Methods: […]. We used ANCOVA models and LOESS regression to examine the associations across gradients of average daily temperature and daily hours of sunlight with daily steps.”
Introduction section:
“This study aimed to examine the associations across gradients of average daily temperature and sunlight hours with daily steps across a 35-week period in school-aged children and to evaluate whether there were differences by sex.”
Comments 5:
L26 – between t avagere daily temperature?
Response 5:
Thank you for the reviewer´s comment. We have corrected the sentence.
Comments 6:
Introduction: The research problem is clearly stated and defined. It is appropriately contextualized in relation to existing knowledge, moving from the general to the specific. The reasons justifying the need for the study are well established in the text. The references used to support the presentation of the research problem are current and relevant to the topic. The objective is clearly stated.
Response 6:
We appreciate the reviewer's perspective on our introduction.
Comments 7:
L74 – “the dose-response”
Response 7:
We appreciate the reviewer's thoughtful consideration of the terminology employed in our study, given that it is a descriptive study. In accordance with the reviewer's suggestion, the term 'dose-response analysis' has been modified throughout the manuscript.
Abstract section:
“Objective: To examine the associations across gradients of average daily temperature and sunlight hours with daily steps across […].”
“Methods: […]. We used ANCOVA models and LOESS regression to examine the associations across gradients of average daily temperature and daily hours of sunlight with daily steps.”
Introduction section:
“This study aimed to examine the associations across gradients of average daily temperature and sunlight hours with daily steps across a 35-week period in school-aged children and to evaluate whether there were differences by sex.”
Comments 8:
Materials and Methods: The methodological procedures are generally appropriate to the study of the research question and are sufficiently detailed and described in an adequate, clear and objective manner.
Response 8:
Thanks to the reviewer for his/her thoughtful comment.
Comments 9:
L146-152 – Why did you categorize daily sunlight hours tertiles and daily temperature tertiles?
Response 9:
We appreciate the reviewer's question. The decision to categorise sunlight hours and temperature into tertiles was made in order to explore potential differences in physical activity between low, medium, and high categories, as well as the identification of non-linear trends and possible thresholds.
Comments 10:
L153 -155 - This explanation about the adjustment of model 0 and model 1 is confusing
Response 10:
The reviewer’s comment seems judicious. We have modified the text to clarify the explanation.
Methods section, 2.4. Statistical analysis:
“In Model 0, no adjustments were made, whereas in Model 1, we controlled for daily sunlight hours and average daily temperature, respectively.”
Comments 11:
Results: Presents the results in an appropriate manner, highlighting the main findings and avoiding unnecessary repetition. Make appropriate use of tables and figures and facilitate appropriate dissemination of results.
Response 11:
Thank you for the reviewer´s comment.
Comments 12:
Table 1. - it doesn't seem appropriate to me to say that it is “Average weekly temperature (ºC) exposure” and “Daily sunlight hours exposure”. The word exposure sounds as if everyone was exposed to it in the same way, which is impossible, even for social reasons, preferences, social commitments, school... even though the week and day had such climatic conditions. I suggest just eliminating the word "exposure"
Response 12:
Thank you for the reviewer´s comment. As suggested by the reviewer, we have deleted the word “exposure”.
Comments 13:
L216- “no significant difference was observed between 10-12 hours and > 12 hours of daily sunlight for boys (model 0).” However, Table 3 Superscript letter indicates statistical significance?
Response 13:
Thank you for the reviewer's comment. In boys, the superscript letter ‘L’ is shown in the medium (10-12 hours) and high (> 12 hours) categories, which means that these categories have significant differences with the Low category (< 12 hours), but not differences were found between medium and high categories.
Comments 14:
Discussion Presents a discussion of the limitations of the study and of existing knowledge on the topic. The strengths of the study are presented and discussed, suggesting the potential contributions of the main findings of the study to scientific development, innovation, or intervention in practice.
Response 14:
Thanks to the reviewer for his thoughtful comment.
Comments 15:
The level of physical activity is multifactorial, and in the case of children, the school period (vacation days or classes) will have a direct influence. You mentioned something in lines 243 and 270-272. This is a factor that should be considered as a covariate in future studies. But it does not invalidate this study. I just think it should be further explored in the discussions and explored in the limitations of the study. Additionally, the methodology, and figures 1 and S3, could somehow indicate the weeks in which the children were in school or on vacation weeks.
Response 15:
Thank you for the comment. To address this issue, we have added the following information:
Methods section:
“Data collection began in October 2022 (yearly week 41) and ended in June 2023 (yearly week 23), lasting 35 weeks. Christmas holidays corresponded to week 52 and 1 and Easter Holidays corresponded to week 14.”
Discussion section:
“However, there are notable differences between a child's typical daily routine during the academic year and that of non-school days, including weekends and holidays. Previous research suggests that structured environments, such as school days, are associated with higher levels of PA due to the inclusion of scheduled physical education classes, recess periods, and after-school activities. In contrast, non-school days may be characterised as a less formally structured environment [35].”
Limitations and strengths section:
“Fourthly, the long-term data collection period included school and holiday periods, which could lead to variability in daily step counts due to changes in children's daily schedules. It would be beneficial for future studies to consider school and holiday periods as covariates to accurately capture these influences.”
Figures:
“Figure 1. Scatterplots illustrating LOESS regression analysis between weeks of the year (from week 41 2022 to week 23 2023) and daily steps, average daily temperature (ºC), and daily sunlight hours. Complete n (n = 655): Week 49 to Week 17. Christmas holidays correspond to week 52 and 1. Easter Holidays correspond to week 14.”
Supplementary material:
“Figure S3. Scatterplots illustrating LOESS regression analysis between weeks of the year (from week 41 2022 to week 23 2023) and daily steps, average daily temperature (ºC), and daily sunlight hours, by sex. Complete n (n girls = 333, n boys = 322): Week 49 to Week 17. Christmas holidays correspond to week 52 and 1. Easter Holidays correspond to week 14.”
Comments 16:
Supplementary material: Good quality material that facilitates the graphical presentation of the main findings of the study.
Response 16:
Thank you for the reviewer´s comment.
Comments 17:
References: The references are updated and sufficient, most of them are made up of references from original articles and the citations in the text are adequate, that is, the statements in the text cite references that substantiate such statements.
Response 17:
We appreciate the reviewer's comments.
Round 2
Reviewer 1 Report
Comments and Suggestions for Authors
The authors answered most of the comments. However, there are some issues which are not being addressed by the author properly, and also some additional issues which have to be addressed.
1. The term “e-MOVI’ is not commonly used.This has to be defined, and it is highly recommended to remove it from the title to avoid confusion
2. I didn’t understand the novelty of this study. There are several studies available in the literature that address the association of sunlight, temperature, and weather with physical activities in children. Some of these studies used more reliable methods of outcome measurement. You have to address how your study varies from these studies. What advancement of knowledge will be there based on this study
3. The authors are explaining the PA trends in some selected countries countries . what is the is the difference between these countries in terms of PA and environment ?
4. Calibration of the device should be included in the methodology
5. The self-reported data, especially in young children is a major flaw of the study
Author Response
Comments 1:
The authors answered most of the comments. However, there are some issues which are not being addressed by the author properly, and also some additional issues which have to be addressed.
Response 1:
We appreciate the reviewer´s effort and comments on the manuscript.
Comments 2:
- The term “e-MOVI’ is not commonly used. This has to be defined, and it is highly recommended to remove it from the title to avoid confusion
Response 2:
Thank you for the reviewer's comment. We have removed “findings from the e-MOVI study” from the title to prevent confusion and have clarified the definition in the methods section:
Title: “Association between temperature and sunlight hours with daily steps in school-aged children over 35 weeks”
Methods section, 2.1. Study design and participants:
”[…] we performed prospective longitudinal analyses of data from the e-MOVI, a study in which PA, dietary behaviour, lifestyle, and cardiovascular risk were evaluated during childhood.”
Comments 3:
- I didn’t understand the novelty of this study. There are several studies available in the literature that address the association of sunlight, temperature, and weather with physical activities in children. Some of these studies used more reliable methods of outcome measurement. You have to address how your study varies from these studies. What advancement of knowledge will be there based on this study
Response 3:
We appreciate the reviewer's comment.
The novelty of our study is mainly based on two aspects. On the one hand, although there are other reliable tools to assess physical activity, the use of steps/day has been recognised as an quantifiable and easy measure for the population. On the other hand, our study examines the influence of environmental factors (sunlight hours and temperature) on physical activity measured by steps over a longer period, whereas previous studies have done so over short periods of time. We have therefore extended the introduction section to support this, as follows.
“Although there are other reliable tools for measuring PA, daily step counts have emerged as a practical and easily quantifiable metric for assessing children's PA levels in both research and real-world settings [17,18]. The daily step count provides a comprehensive representation of the overall PA accumulated throughout the day. This is a crucial metric in studies examining the relationship between PA and environmental factors, as it offers a comprehensive insight into the PA patterns of an individual over a specified period. However, few studies have employed daily steps as an objective measure of PA [19,20].”
“Furthermore, most studies analysing the determinants of PA, particularly daily steps, have been conducted over relatively short periods of time, often assessing PA for a period of seven days or less or comparing specific seasonal periods, which could contribute to bias because the data are collected in a restricted period of the year. Thus, there is limited evidence on how daily variations in temperature and sunlight hours impact daily steps in children over a prolonged period.”
Comments 4:
- The authors are explaining the PA trends in some selected countries countries . what is the is the difference between these countries in terms of PA and environment ?
Response 4:
Thank you for the comment. As the reviewer suggests, we have added some extra information in the introduction section to explain the differences between the mentioned countries in terms of physical activity and environment, as follows:
“For example, children in colder climates in winter, such as northeast America and northeast Spain, may have greater difficulties in engaging in outdoor activities during this season, whereas those in regions with mild winters and warm summers, such as southeast Australia and southwest France may have more opportunities for engagement throughout the year [13].”
Comments 5:
Calibration of the device should be included in the methodology
Response 5:
Thank you for the comment. As the reviewer suggests, we have added the explanation of calibration device in the methodology, as follows:
Methods section, 2.3.2. Outcome: daily steps
“Before initiating the study, the accuracy of the step-counting device was assessed on a subsample of the participants. This pilot calibration test was performed on a treadmill using a protocol that required participants to complete up to 10 sets lasting 3-min at 0% incline. Treadmill speed was started at 0.8 km/h and increased 0.8 km/h for each set. Participants were not given specific instructions on the type of locomotion to perform. The protocol ended when either the participant or the investigator decided to terminate the test. The accuracy of the device was assessed by comparing the recorded step count with a manual count. In addition, a video camera focused on the feet of each participant was used to obtain a secondary record. The intraclass correlation coefficient obtained was 0.93 between the step count recorded manually by direct observation and the count recorded by the Xiaomi Mi Band 3, indicating high agreement between the two methods.”
Comments 6:
The self-reported data, especially in young children is a major flaw of the study.
Response 6:
Thank you for the comment. We have reinforced this limitation, as follows:
Limitations and strengths section:
“Finally, the data on daily steps were self-reported by the children, which could introduce the potential for reporting bias. However, we were unable to access the direct data from the schoolchildren due to Xiaomi Inc.’s privacy policy, preventing access to the data. Thus, we endeavoured to minimise bias and enhance reliability by implementing a rigorous data collection procedure and providing clear instructions to the participants. Future studies should consider using objective measures to improve data reliability.”
We thank the reviewer for his/her relevant contributions. In our opinion, we believe that the article has been considerably improved. If there are any specific aspects of the manuscript that the reviewers and editor believe still require attention or if there are additional suggestions for improvement, we welcome further feedback to ensure the quality and compliance of our work with the journal's guidelines.
